# Developmental Trajectories of Transcription and Oral Language Skills in Kindergarten Students: The Influence of Executive Functions and Home Literacy Practices

**DOI:** 10.3390/jintelligence13120163

**Published:** 2025-12-13

**Authors:** Jennifer Balade, Cristina Rodríguez, Juan E. Jiménez

**Affiliations:** 1Department of Developmental and Educational Psychology, Faculty of Psychology and Speech Therapy, Campus Guajara, Universidad de La Laguna, 38200 San Cristóbal de La Laguna, Spain; ejimenez@ull.edu.es; 2Millennium Nucleus for the Science of Learning (MiNSoL), Linares 3582979, Chile; mrodriguez@ucsc.cl; 3Facultad de Educación, Universidad Católica de la Santísima Concepción, Concepción 4090541, Chile

**Keywords:** writing development, kindergarten, executive functions, home literacy practices

## Abstract

This study investigates the developmental trajectories of transcription and oral language skills in kindergarten students over the course of an academic year, with a focus on the influence of executive functions (EF) and home literacy practices (HLP). Hierarchical linear modeling (HLM) analyses revealed significant growth in transcription skills, with both EF and independent home literacy practices positively influencing baseline transcription scores. The interaction between independent home literacy practices and formal literacy practices at home further enhanced transcription skill development. In contrast, oral language skills were not influenced by either HLP or EF. These results suggest that EF plays a more prominent role in transcription development than oral language skills in early childhood, especially in transparent orthographic systems. The findings highlight the importance of cognitive and environmental factors in early literacy development, suggesting implications for educational practices, particularly in fostering effective home literacy environments

## 1. Introduction

Proficiency in writing is a multifaceted skill crucial for academic and professional success ([83]). Its development is shaped by the interplay of several key dimensions, including cognitive skills (e.g., executive functions), linguistic abilities, and social literacy practices ([5]; [98]). The influence of these dimensions varies with developmental stage, contributing to the individual differences in writing performance observed even within the same classroom. Furthermore, these dimensions do not exert influence simultaneously or uniformly. Depending on factors such as age and neurological maturation, certain domains may play a more prominent role than others at different stages of development. Additionally, each dimension evolves uniquely based on individual experiences and abilities, which are not consistent across all students ([98]). As a result, varying levels of writing performance can be observed within the same classroom environment.

In the early stages of writing development, mastering the foundational skills of transcription and oral language is a cognitively demanding process that is crucial for later, more advanced composition abilities ([17]; [73]; [101]). This cognitive demand highlights the potential role of child-specific abilities like executive functions (EF) and environmental influences such as home literacy practices (HLP) in explaining the pronounced individual differences in early writing. Gaining a deeper understanding of how these factors contribute to early writing skills is crucial for identifying the mechanisms that support or hinder writing acquisition. This study aims to examine the developmental progression of transcription and oral language skills in early grades over the course of a school year, investigating how executive function abilities and home literacy practices may influence the growth of these skills. 

### 1.1. Writing Development at Early Grades

During kindergarten (ages 5–6), children’s writing undergoes rapid development, with significant gains in foundational transcription skills (e.g., handwriting, spelling) and oral language complexity ([16]; [26]; [107]). These parallel advances in transcription and oral language, moving from idiosyncratic productions to increasingly conventional discourse, supported by growing phonological awareness and other metalinguistic skills ([26]; [107]). Several studies in kindergarten populations have documented improvements in letter formation and handwriting fluency across the school year ([36]; [50]), as well as growth in vocabulary and syntactic complexity in oral expression ([35]). These foundational skills provide the basis for early transcription and text generation processes, which are further shaped by cognitive and environmental factors such as executive functions and home literacy practices.

Recent findings by [93] ([93]) advance this understanding by demonstrating that transcription skills moderate the relationship between oral language and writing quality in kindergarten writers. Specifically, their study found that for children with low transcription skills, stronger oral language abilities can partially compensate and positively influence writing quality. Conversely, for children with average transcription skills, oral language shows little effect, while surprisingly, among those with high transcription skills, lower oral language ability was associated with better writing quality. This counterintuitive result may reflect that children with stronger oral language attempt more complex writing tasks that exceed their transcription capabilities, thereby reducing their quality scores due to transcription errors. This evidence supports the notion that transcription demands in early writers may overshadow oral language influences, particularly regarding writing quality, while the contributions of oral language remain critical and should not be neglected in instructional contexts.

### 1.2. Theorical Models of Writing

Several theorical models have been developed to understand the writing development. The Simple View of Writing ([14]) highlights transcriptions and text generation as critical skills to mastering writing process. Its extension, the Not-So-Simple View of Writing (NSVW; [17]), integrates higher-order cognitive components, namely executive functions, as essential processes that interact with both transcription and text generation. Similarly, the Direct and Indirect Effects model of Writing (DIEW; [74]; [73]) underscores the reciprocal influence of transcription, oral language, and executive functions in early writing development. Research highlights the importance of these cognitive capacities, including executive functioning, as key dimensions of writing performance ([5]; [48], [49]; [79]).

According to the recently proposed DIEW model ([73]), the main writing components of early writing include transcription and oral language skills. In this framework, transcription and oral language skills still play a crucial role in written composition. Moreover, the authors maintain that executive functions are primarily directed toward transcription in the early stages, and as this skill becomes more automatic, cognitive resources are freed for higher-order skills such as oral language discourse ([73]), also aligning with the theories of DCH ([12]). Transcription consists of handwriting and spelling, as well as phonological and semantic domains. In contrast, oral language discourse represents the foundation of text generation processes. Both transcription and oral language discourse exert a reciprocal influence on each other, and they are also influenced by executive functions, which include attention, working memory, inhibitory control, and cognitive flexibility.

According to these models, transcription is understood as the ability to translate ideas into written symbols, relying on working memory ([13]; [51]). This skill encompasses both handwriting and spelling. Handwriting not only requires the sequencing of finger movements but also requires access to the shapes of letters and words stored in working memory, so it is not considered a motor sub-skill in its entirety, but also a linguistic one ([14]). Moreover, handwriting fluency positively affects spelling and phonological skills at early grades ([96]). On the other hand, spelling requires retrieving words from memory to access their orthographic patterns ([33]). These two subskills, while distinct, are highly interrelated components essential for writing development ([14]; [63]; [100]). Furthermore, spelling relies on phonological awareness ([4]; [40]) an essential skill for accurately encoding words through both orthographic and phonological processes ([12], [15]). Additionally, a strong relationship exists between lexical knowledge and spelling proficiency. A well-developed and extensive vocabulary enhances spelling accuracy ([2]; [62]) Additionally, transcription is classified as a low-level skill within theoretical models because, during the initial stages of writing acquisition, it demands significant cognitive resources such as attention and working memory. However, as transcription becomes automatized, these cognitive resources are gradually freed, allowing for greater engagement in higher-level writing processes, such as text generation ([17]; [63]).

Building on these models, text generation (or ideation) refers to the ability to transform oral ideas into written texts, a process strongly grounded in oral language discourse ([17]). Strong oral language skills are closely related to writing proficiency, particularly in the early stages of learning to write. This connection stems from parallel and interactive development of writing and metalinguistic skills across various levels ([106]), with oral language significantly supporting early writing development. Furthermore, prior work validated the structural model of writing for Spanish-speaking children, confirmed that transcription and oral language function as two distinct yet highly interrelated factors ([65]; [101]). These findings, derived from structural equation modeling, underscore the critical role of oral language in the writing construct. This distinction is particularly critical within the context of shallow orthographies like Spanish. A shallow orthography is characterized by a high degree of consistency between graphemes (letters) and phonemes (sounds), which contrasts with opaque orthographies like English ([107]).

The nature of a shallow orthography is crucial. It is theorized that the consistency of Spanish may allow children to master transcription skills with less cognitive load and at a faster pace. This, in turn, could have significant implications for the role of EF and the timeline in which cognitive resources are freed up to support higher-level text generation, which relies on oral language skills ([10]). Therefore, examining these developmental trajectories in a shallow orthography provides a crucial point of comparison to the bulk of research conducted in opaque systems.

### 1.3. Influence of Executive Functions on Writing Development

According to [5] ([5]), there are different essential dimensions that influence writing development. One of them is cognitive abilities, referring to executive functions. Extensive research has demonstrated the impact of executive functions on writing ([102]). However, executive functions have been categorized into two levels ([34]): (1) low-level executive functions, involving substantial attention, working memory, inhibition, and cognitive flexibility; and (2) high-level executive functions, which involve reasoning, planning, and problem-solving.

Regarding low-level executive functions, attention has been found to play a central role in handwriting skills in kindergarten children ([90]). Working memory provides the ability to hold and manipulate the necessary information needed to complete a writing task and, additionally, retrieve information from long-term memory and integrate it into a temporarily held mental representation ([15]; [53]; [116]). Working memory is particularly relevant for spelling and is also linked to oral language development and cognitive flexibility in young children ([42]; [68]; [121]). Inhibition allows writers to select relevant information while ignoring irrelevant details in order to compose longer, more cohesive, and more accurately spelled texts ([3]; [27]; [38], [39]; [60]; [99]). Specifically, inhibitory control has been associated with spelling accuracy and indirectly with text generation through transcription, especially handwriting ([29]; [68]; [103]; [121]). Cognitive flexibility involves adapting quickly to changing task demands and has been shown to influence spelling in kindergarten children ([34]; [114]). In addition, oral language skills in kindergarten children have been shown to relate more closely to verbal working memory and cognitive flexibility than to inhibition or visual working memory ([42]), highlighting the role of these executive functions in supporting early text generation and the integration of oral ideas into written form.

Concerning high-level executive functions, reasoning refers to the capacity to generate innovative ideas, anticipate actions, and use effective and strategic approaches. Planning encompasses formulating ideas and setting objectives, transforming mental representations into a structured sequence of actions and strategies to compose structured and cohesive texts. Problem-solving involves identifying the most effective solution to address a given challenge ([34]).

At the beginning of writing development, younger students focus their cognitive resources on mechanical and local problems ([83]) rather than on planning, reviewing, or correcting their writing output ([80]). In addition, the mastery of low-level executive functions supports the development of high-level executive functions ([34]). A systematic review carried out by [102] ([102]) examined the contribution of executive functions to children’s writing. A total of 26 studies met the inclusion criteria. The participants across the included studies ranged from 6 to 18 years old, which indicates that findings primarily concern school-aged children and adolescents. The results showed that working memory, inhibition, cognitive flexibility, and planning were the components studied to date—with a special focus on working memory. All these components showed a significant relationship with writing skills across different tasks and dimensions of writing (e.g., quality and productivity). The authors concluded that these components support writing skills at early ages. However, they also noted that there is limited research on this topic, particularly regarding shallow orthographies, as most studies in the review were conducted in opaque orthographies. They also highlighted the importance of considering the orthographic system when designing studies, as other authors have mentioned ([81]). This gap further underscores the need to explore these relationships in preschool children, such as those included in the present study.

It is also important to consider the developmental nature of executive functions when studying their influence on early writing skills. During the early childhood years, particularly from ages 3 to 8, executive functions operate more as a unified construct rather than as clearly differentiated components ([23]). This unitary functioning gradually evolves into three interrelated but distinct components as children grow older ([86]). Therefore, when working with kindergarten students, it may be more appropriate to conceptualize and measure executive functions as a holistic construct, rather than assessing isolated components such as working memory or inhibition separately. This approach aligns with recent recommendations in literature and supports more developmentally sensitive modeling of executive functions in early writing research ([23]; [39]; [60]; [86]; [115]).

In a recent study, [103] ([103]) examined the contribution of executive functions to transcription skills. Their findings revealed that executive functions significantly influenced both handwriting and spelling with a stronger effect on spelling. Although this study advances our understanding of executive functions’ role in transcription, it did not explore their impact on oral language skills.

### 1.4. Influence of Dependent and Independent Literacy Practices on Writing Development

While domain-general cognitive factors, such as executive functions, significantly influence writing skills development, external environmental factors also play a critical role. According to Bronfenbrenner’s Ecological Systems Theory, family and school contexts form the foundation of child development ([22]). Just as teachers provide opportunities to develop writing skills at school, parents foster literacy through home literacy practices ([30]). Extensive research has shown that HLP play a pivotal role in early childhood literacy development ([56]; [87]; [104]), and this early engagement with literacy may significantly influence students’ later writing performance ([98]).

HLP which can be categorized into two distinct forms: dependent and independent practices. Dependent HLP involve literacy activities facilitated by an external agent, such as a family member actively participating in the task. These practices are further divided into dependent-formal practices characterized by structured, explicit interactions with written language, and dependent-informal-practices types, which occur incidentally through everyday activities guided by the agent ([54]; [57]; [58]; [94]). In contrast, independent HLP encompass children’s self-initiated literacy activities without adult support or guidance. This autonomy-supportive approach aligns with Self-Determination Theory, fostering intrinsic motivation by encouraging children to engage in writing for personal satisfaction rather than external rewards ([31]).

Research suggests that dependent-formal HLP, such as structured writing activities guided by parents, predict handwriting, spelling, and spontaneous writing skills in early childhood ([1]; [7]; [25]; [94]). These effects may persist in the short and medium term ([54]) and even into the long term ([6]; [8]; [78]). Similarly, independent HLP, involving self-initiated writing activities by the child without parental guidance, also positively influence spelling ([7]; [54]), handwriting ([54]; [94]), and spontaneous writing ([94]). Dependent-informal HLP, such as incidental literacy exposure, enhance children’s oral language skills ([24]; [105]) but show no significant impact on writing performance.

To date, and to the best of our knowledge, no studies have examined how the development of foundational writing skills—specifically transcription and oral language—evolves over time during the final year of early childhood education. Moreover, there is a lack of research exploring how external influences, such as home literacy practices or children’s executive function skills, may shape the growth trajectories of these foundational abilities throughout this critical period.

### 1.5. The Present Study

A growing body of evidence highlights the importance of considering the combined influence of home literacy practices and executive functions on children’s literacy development. Home literacy practices have been shown to be positively associated not only with early writing skills but also with the development of executive functions during early childhood ([77]; [108]). Executive functions, in turn, play a fundamental role in writing performance due to their involvement in planning, self-regulation, and working memory processes ([11]; [17]; [20]; [73]).

This study pursues two primary objectives: (1) to examine the growth trajectory of transcription skills and oral language abilities over the course of a kindergarten academic year; and (2) to investigate whether transcription and oral language, along with their growth trajectories, are influenced by internal cognitive processes, such as executive function skills, and external factors, such as formal, informal, and independent home literacy practices. In addition, socioeconomic status, gender and age were included as a control variable in the analysis.

## 2. Materials and Methods

### 2.1. Participants

The sample consisted of 130 kindergarten students (61 boys and 69 girls) from seven schools in the province of Santa Cruz de Tenerife, Spain. These schools included state, semi-private, and private institutions, located across different areas of the island, representing diverse socioeconomic backgrounds. Students with a diagnosed special educational need, such as autism spectrum disorder, Down syndrome, or intellectual, sensory, and/or motor disabilities, were excluded from the sample. This information was provided by the institution. The average age of students at the start of the study was 5.38 (boys: 5.41, SD = 0.30; girls: 5.35, SD = 0.29). Before commencing the study, a letter was sent to parents requesting their informed consent. The schools distributed this letter, and only students whose parents returned the signed consent form were included in the study. Once this process was completed, the study proceeded.

### 2.2. Procedure

The study adhered to the ethical standards established by the Research Ethics Committee of the *Universidad de La Laguna*, including the provision of an informed consent form that was distributed to the parents of all participating students. This study began in October 2021 with assessments of transcription and oral language skills, repeated in February and May 2022. Executive functions and home literacy practices were assessed once in May 2022. Transcription and oral language were evaluated using the T-IPAE-K digital tool on tablets, designed to identify kindergarteners at risk for writing difficulties ([65]). This tool includes tasks in three forms (A, B, C) administered at the start, mid, and end of the school year, showing strong validity and diagnostic accuracy ([65]). Executive function tasks, adapted from standardized tests, were also tablet-based, while home literacy practices were assessed via an online questionnaire distributed to families, capturing socioeconomic status and formal, informal, and independent literacy practices. Eighteen graduate student examiners from speech therapy, psychology, or education fields administered and scored the measures after two training sessions per assessment period, covering task administration, app use, and scoring. Real-time scoring for tasks like phoneme isolation, expressive vocabulary, and executive functions used an external keyboard to log responses instantly.

### 2.3. Instruments and Measures

#### 2.3.1. Transcription

Transcription was assessed through handwriting (letter copying task) and spelling (name writing task), complemented by phoneme isolation and expressive vocabulary, which are well-established predictors of these skills ([4]; [40]; [62]). This structure has been previously validated in related studies ([65]).

Letter copying. This task measured children’s handwriting fluency by evaluating their ability to accurately reproduce letters within a one-minute timeframe. This task has been widely employed to assess handwriting automaticity in children ([67]; [75]; [117]). Children were asked to copy the five vowels presented sequentially on a screen, one at a time. Vowels were specifically chosen because, unlike consonants, their instruction is consistently guaranteed across all kindergarten curricula in our educational system, ensuring comparability among participants despite heterogeneity in early literacy practices across schools. The final score represented the total number of correctly reproduced letters within the given time. A letter was considered correct if it showed no errors such as omissions, misalignments, additions or reversals. The intraclass correlation coefficient (ICC) for this task was 0.960, with 95% confident intervals ranging from 0.948 to 0.99 (F (214, 214) = 49.2, *p* < .001). Once the task was completed, the examiners reviewed and scored it afterward.

Name writing. This task has been widely examined in previous research ([46]; [112]). Name writing task is comprised of two parts. In the first part, children were asked to write their own name. In the second part, they were instructed to write the names of friends, classmates, family members, or other familiar individuals. After completing the second part, children were asked to verbally identify what they had written on each line. The total number of correctly written words was recorded in the second part. A word was considered correct only if it included all the required letters. Spelling variations involving homophones (e.g., Eva vs. Eba) were not penalized, nor was the use of uppercase versus lowercase letters. To ensure scoring consistency, we also assessed interrater reliability in a random subsample of 117 children, independently scored by two examiners. The ICC was 0.955, with 95% confidence intervals ranging from 0.935 to 0.968 (F (115, 115) = 43.3, *p* < .001), indicating excellent reliability.

Phoneme Isolate. This task aimed to evaluate children’s ability to isolate the initial phoneme of spoken words, an essential phonemic awareness ability for children beginning to read in languages with shallow orthographies ([82]; [84]). Children were instructed to listen to a word and produce its first sound. Each word appeared one at a time. Additionally, children could listen to the word again if needed; however, this repetition was allowed only once per item. Examiners recorded responses, logging correct and incorrect answers in real time using an external keyboard. The task consisted of 18 words. Each measurement session included different items. Each correctly identified phoneme was awarded two points, while responses providing the corresponding letter name received one point. The maximum score was 36. The reliability of this measure, as indicated by Cronbach’s alpha, was 0.96.

Expressive Vocabulary. This task aimed to assess children’s lexical knowledge, given its close relationship with the orthographic buffer—a key component in spelling ([2]; [18]; [43]; [62]). Children were shown a series of images, one at a time, on a tablet screen and were asked to name each image aloud. Examiners recorded responses, marking correct and incorrect answers using an external keyboard. Synonyms or alternative correct labels for the target image were accepted as correct responses (e.g., both “mountain” and “hill” were scored as correct). This approach ensured that children’s lexical knowledge was accurately captured, without penalizing them for using valid alternative words. Each measurement included a total of 10 items. The Cronbach’s alpha for this task was 0.85. This task did not require post-task correction, as examiners recorded hits and misses in real time using the external keyboard.

#### 2.3.2. Oral Language

Oral narrative story. An oral story production task was used to assess children’s narrative competence, a method previously employed in studies with young children ([92], [91]). Children were instructed to create a story based on a provided a different oral prompt for each measurement taking. These prompts were: “One day I wake up and I am invisible” for the October measurement; “One day I wake up and I can breathe underwater” for February measurement; and “One day I wake up and I can fly” for the last measurement taking in May. Before starting the task, the app’s virtual assistant provides the child with 30 s to reflect and organize their thoughts. Once this preparation time ends, the task begins, and the child starts narrating their story. The tablet records the child’s performance, and after the task is completed, the examiner reviews the recording to assess and score the narrative, integrating the components described below.

The assessment focused on three key components: (1) Number of Unique Words (UW), which captured lexical diversity by counting the total number of distinct words in the child’s narrative. Each non-repetitive word was assigned 1 point, and the final score was the sum of all unique words produced; (2) Total Number of T-Units (TU), which assessed syntactic complexity, following the definition by [52] ([52]), where a T-Unit consists of an independent clause along with any associated subordinate clauses. The final score represented the total number of T-Units in the child’s oral narrative; (3) Total Number of Correct Sequences (CS), which evaluated grammatical accuracy by counting pairs of consecutive words that formed a grammatically plausible sequence in the target language.

The reported ICC values for UW, TU, and CS include this blind double-rating and have been partially published in prior studies ([64]; [65]). For the UW, the average ICC was 0.96, with a 95% confidence interval between 0.94 and 0.97, showing statistical significance, F (114, 115) = 23.0, *p* < .001. For TU the average ICC was 0.85, with a 95% confidence interval ranging from 0.79 to 0.89, also demonstrating statistical significance, F (114, 115) = 6.6, *p* < .001 ([64]). Finally, CS obtained an average ICC of 0.95, with a 95% confidence interval between 0.95 and 0.96, showing statistical significance, F (354, 354) = 20.56, *p* < .001 ([65]).

#### 2.3.3. Executive Function

Perception of differences test (*Caras-R*) ([113]). This task evaluated attentional processes through 60 sets of items. Each item consisted of three illustrated faces: two identical and one differing by a subtle detail (e.g., open mouth vs. closed mouth). Children were instructed to identify and round with a pen the different face that differed from the others for each item. This task was the only executive function assessment conducted using a traditional paper-and-pencil format. After the children completed the task, examiners scored the task. The final score was based on the total number of correct responses completed within a 3-min timeframe. According to the test manual, the test-retest reliability was reported as 0.60, while the split-half reliability was 0.97.

Inhibitory control ([9]). This task, adapted from the Stroop day–night paradigm ([47]), measured children’s ability to suppress automatic responses based on semantic content while accurately processing visual stimuli. It comprised two main blocks using images of suns and moons. The first block represented the congruent condition, where children were instructed to say “sun” when shown a sun and “moon” when shown a moon. The second block, the incongruent condition, required children to reverse their responses, saying “sun” when presented with a moon and “moon” when presented with a sun. Each block included 50 items, displayed in a 5-row by 10-column matrix. Children were asked to respond as quickly and accurately as possible within a 45-s time limit for both conditions. At the end of the task, the examiner recorded the total number of correct responses and the last item completed by interacting with the tablet. This allowed them to accurately input the final score on the tablet afterward. The final score was calculated by dividing the total number of correct responses by the time taken during the incongruent block. The test-retest reliability for the incongruent condition was reported as 0.91.

Cognitive Flexibility. This task, adapted from the Dimension Change Card Sort Task ([120]), evaluated cognitive flexibility, which refers to the ability to adapt behavior in response to changing rules or instructions. Children were shown two images at the bottom of the screen: a blue cat and a red car, while target images appeared at the top. The task comprised three blocks: (1) Pre-change: Children categorized the target images based on either color or shape. The starting dimension was counterbalanced—half began with color and half with shape. In the color condition, children matched a blue car to the blue cat and a red cat to the red car. In the shape condition, they matched a red cat to the blue cat and a blue car to the red car; (2) Post-change: Children switched to the opposite dimension—those who started with color now sorted by shape, and vice versa; (3) Border: Target images appeared either with or without a black border. Children followed one rule when a border was present (e.g., sort by color) and another rule when no border appeared (e.g., sort by shape), or the reverse, depending on their initial condition. The task included six items in the pre-change block, six in the post-change block, and 14 in the border block. The performance score was a fluency index, calculated by dividing the total number of correct responses by the time taken in the border block, the most cognitively demanding phase. The task’s reliability, measured by Cronbach’s alpha, was 0.80. This task did not require post-task correction, as the software recorded hits and misses in real time through the images that children click on

Digit span Backward. Adapted from the WISC-V digit span task ([118]), this measure assessed students’ working memory. It consisted of two blocks: forward and backward. Only the backward block was the final score used in this study. In this block, students listened to a sequence of numbers and were required to repeat the sequence in reverse order (e.g., hearing 1–7 and responding with 7–1). The sequence length increased after every four sequences, with a total of 18 sequences presented. The final score was the last full span that came in. The Cronbach’s alpha for this sample was 0.78. This task did not require post-task correction, as examiners recorded hits and misses in real time using the external keyboard.

Oral cloze task. This task, adapted from the work of Siegel and Ryan ([109]), assessed verbal working memory. The examiner orally presented a series of incomplete sentences, each missing the final word. Students were instructed to fill in the missing word and later recall the missing words in the exact order the sentences were presented. Before starting, students completed a practice trial consisting of two sentences, with two attempts allowed for this practice set. The main task included six sets, each allowing three attempts. The first set contained two sentences, gradually increasing to seven sentences in the sixth set. The task ended if the student failed all attempts within a set. The final score reflected the total number of successfully recalled attempts, with a maximum possible score of 18 points. The task demonstrated a reliability index of 0.73, as indicated by Cronbach’s alpha. This task did not require post-task corrections, because examiners recorded hits and misses responses in real time through the external keyboard.

#### 2.3.4. Parents’ Questionnaire: Home Literacy Practices and Socio-Economic Status

Parents completed an online survey accessed via a QR code. The survey was divided into two sections: the first collected information on socioeconomic status, while the second focused on home literacy practices.

Socio-economic status (SES). The socio-economic status section was also divided into two different categories: Educational level and income level. Educational level was scored from 0 to 5: no formal education (0), primary education (1), secondary education or intermediate vocational training (2), advanced vocational training (3), university degree (4), postgraduate or higher (5). Income level was scored from 0 to 4 based on monthly household contributions: no income (0), 3500€ (4). Each parent received scores for both metrics.

Previous research suggests that the most effective way to measure a family’s socioeconomic status (SES) is by considering both income and educational level ([21]). Following this approach, the SES score was calculated as the sum of the scores obtained from both parents across the education and income sections ([28]). This method represents a compensatory weighted composite, where a higher score in one domain (e.g., income or education) can partially offset a lower score in the other ([54]). Therefore, the maximum score for SES was 18.

Home Literacy Practices (HLP). The home literacy practices (HLP) section was organized into three categories: dependent-formal (DF_HLP), dependent-informal (DI_HLP), and independent practices (I_HLP). This classification follows previous research that distinguishes dependent and independent practices, with dependent practices further divided into formal and informal types (e.g., [54]; [88]; [94]; [108]). Parents rated activity frequency on a scale from 1 (rarely/never) to 4 (almost daily). DF_HLP included three items: (1) helping the child to write letters, (2) helping the child to write their name, and (3) engaging in joint formal writing activities. DI_HLP consisted of five items: (1) shared drawing or coloring activities, (2) playing rhyming games, (3) manipulative play with letters (plastic, wooden, or magnetic), (4) tracing letters on different surfaces and materials (e.g., sand, chalkboard, playdough, chalk), and (5) using mobile applications for letter-tracing. I_HLP referred to three items involving children’s self-initiated activities without adult support: (1) independently writing letters, (2) independently writing words, and (3) independently engaging in drawing or coloring activities.

### 2.4. Data Analysis

All analyses were conducted in R ([97]). Longitudinal data were transformed into a long format, with each student assigned three rows for October, February, and May measurements, coded by a “time” variable (1, 2, 3). Transcription (TR) and oral language (OL) varied by time, while executive functions (EF) and home literacy practices (HLP) remained constant. TR was computed as the sum of copying letters (CL), name writing (NW), phoneme isolation (PI), and expressive vocabulary (EV) scores; OL was derived from UW, TU, and CS scores, with both structures validated previously ([65]). EF was calculated from perception of differences, inhibitory control, cognitive flexibility, digit span backward, and oral cloze task scores, validated via confirmatory factor analysis (CFA) using fit indices (chi-square, CFI, TLI, RMSEA, SRMR) and maximum likelihood estimation with robust standard errors (MLR; [61]; [66]). For HLP, a network analysis identified item clusters (using degree, eigenvector, closeness, and betweenness centrality), followed by CFA with a Weighted Least Squares Mean (WLSM) estimator to confirm the factor structure ([37]; [59]). A correlation analysis assessed relationships between TR, OL, EF, and HLP. Separate HLMs for TR and OL modeled within-subject and within-school random effects, testing: (1) random effects, (2) a base model, and (3) a final model with significant EF and HLP predictors, selected via stepwise comparison using AIC, log-likelihood, and deviance statistics ([85]; [95]).

As mentioned above, the procedure for HLM involved three main steps. First, intraclass correlations (ICCs) were estimated to evaluate the need for student- and school-level random effects (phase 1). Second, different random-effects structures were compared systematically to identify the most appropriate configuration for each outcome variable (phase 2). Finally, fixed effects representing HLP and EF were added to the baseline random-effects model, and models including fixed effects on the intercept only versus both intercept and slope were evaluated to determine where each predictor exerted its influence (phase 3). This stepwise approach ensures that the final model provides reliable estimates of fixed effects while accounting for hierarchical dependencies in the data, a standard procedure in longitudinal HLM analyses (e.g., [44]; [110]). In cases where significant interaction effects were detected in the final models, we conducted post-hoc probing to determine their direction and nature.

## 3. Results

Descriptive statistics of longitudinal variables for TR and OL are presented in Table 1. Moreover, the descriptive statistics for non-longitudinal variables, such as EF variables or the questionnaire, are reported in Table 2. In both tables, skewness and kurtosis do not present issues with data distribution due to their absolute values do not exceeding the thresholds of 3.0 for skewness and 10.0 for kurtosis, in line with the guidelines suggested by [76] ([76]).

### 3.1. Executive Functions Structure

During CFA analysis for EF, a high Modification Index (MI) between observed variables OC and CF indicated an unaccounted-for relationship. Allowing a correlation between their measurement errors improved model fit without compromising theoretical integrity ([111]). The adjusted EF model showed good internal consistency (Omega = 0.723) and fit: non-significant chi-square (χ^2^ = 3.34, *p* > .05), NFI = 0.96, NNFI = 1.00, CFI = 1.00, TLI = 1.00, SRMR = 0.02, RMSEA = 0.00 ([55]; [69]). EF was computed as the sum of its observed measures.

### 3.2. Home Literacy Practices Structure

The HLP questionnaire’s structure was validated in two phases: (1) network analysis to identify item clusters and (2) CFA to confirm the factor structure. In the network analysis, items were nodes, with edges representing correlations (>0.2). Centrality measures (degree, eigenvector, closeness, betweenness) highlighted DF_3, DI_1, DI_2, DI_4 (degree = 14), and I_2 (betweenness = 7.55) as key nodes. Community detection using the Fast-Greedy algorithm identified two communities: one with dependent-formal (DF) and dependent-informal (DI) practices, showing a substructure of formal/informal items, and another with independent (I) practices ([59]). See Table 3 for centrality and network visuals.

Two CFA models were tested: a two-factor model (dependent, independent practices) and a three-factor model (DF, DI, I practice). The two-factor model required adjustment for high Modification Indices between DF_1 and DF_2, achieving good consistency (Omega = 0.904) but mixed fit: non-significant adjusted chi-square (χ^2^ = 38.83, *p* > .05), CFI = 0.93, RMSEA = 0.07, SRMR = 0.06, NFI = 0.83, NNFI = 0.91, TLI = 0.91 ([55]). The three-factor model, needing no MI adjustment, showed better fit: non-significant chi-square (adjusted: χ^2^ = 28.70, *p* = .93; scaled: χ^2^ = 55.60, *p* = .06), NFI = 0.85, NNFI = 0.94, CFI = 0.96, TLI = 0.94, RMSEA = 0.05, SRMR = 0.05, and Omega = 0.904. Due to its parsimony, the three-factor model was adopted, with DF_HLP, DI_HLP, and I_HLP computed as sums of their items.

### 3.3. Correlation Between Longitudinal and Non-Longitudinal Variables

A correlation analysis was conducted to examine which of the non-longitudinal variables (EF, DF_HLP, DI_HLP, and I_HLP) maintained significant correlations with the longitudinal variables: TR and OL (see Table 4). A significant correlation was found between almost all measures of TR and DF_HLP, I_HLP, and EF. In contrast, DI_HLP and DF_HLP were the only measures that obtained a significant correlation with OL in some moments measured. Based on this result, HLM for TR was performed, considering the presence of EF, dependent-formal, and independent practices, while HLM for OL included the presence of dependent-informal and dependent-formal practices. The SES measure was included in the model as a covariable.

The growth of TR and OL scores across the three time points, without the interaction of fixed and random effects, can be observed in Figure 1.

### 3.4. Hierarchical Linear Modeling for Transcription

To model transcription trajectories, we conducted a Hierarchical Linear Modeling analysis. First, intraclass correlation coefficients (ICCs) confirmed significant variance at both the student (ICC = 0.59) and school (ICC = 0.15) levels, justifying the use of a multilevel approach ([45]). After testing several random effects structures, the most parsimonious baseline model was selected, according to AIC, log-likelihood, and deviance values, residuals and intercept-slope correlation ([41]; [119]). This final model included random intercepts and slopes at the student level, and random intercepts at the school level. The predictors and control variables were then added to this baseline model as fixed effects.

The final model, which incorporated the fixed effects of home literacy practices (DF_HLP, I_HLP), executive functions (EF), and control variables (SES, gender, age). The model with effects on intercept and slope was most parsimonious (χ^2^(11) = 34.98, *p* < .000) and presented lowest AIC, LogLik and deviance, with residuals (−1.92 to 2.62) and correlation (r = 0.70).

Significant intercept effects included SES (β = 0.52, *p* < .01), I_HLP (β = 2.53, *p* < .000), EF (β = 0.72, *p* < .000), and the interaction between I_HLP and EF (β = 0.22, *p* = .02). These results suggest that students’ final transcription performance may be shaped by their socioeconomic background, the extent of independent literacy practices at home, their executive function proficiency, and the interaction between the latter two factors (Table 5).

Significant effects on the growth trajectory of TR were observed for executive functions (β = 0.36, *p* < .001) and independent home literacy practices (β = 0.89, *p* = .01). An additional significant effect on the slope emerged from the interaction between dependent-formal and independent home literacy practices (β = 0.29, *p* = .02). We probed this interaction by plotting the simple slopes of time and I_HLP at low, medium, and high levels of DF_HLP, as shown in Figure 2. The plot illustrates the direction of this effect: For children with high levels of dependent-formal practices (right panel), there is a strong, positive relationship between independent practices and the growth of transcription skills over time (i.e., the growth slope is steepest for children with the highest I_HLP). In contrast, for children with low levels of dependent-formal practices (left panel), the level of independent practices has little to no effect on the growth trajectory. This suggests a synergistic effect, where independent and formal home literacy practices combine to accelerate transcription skill development.

Importantly, beyond the influence of specific predictors, transcription skills showed a significant overall improvement across the three measurement points (β = 6.80, *p* < .001), indicating that children’s TR abilities developed consistently over the course of kindergarten.

### 3.5. Simple Slopes Analyses and Pairwise Comparisons for Transcription

To clarify the nature of the significant interactions found in our model for transcription, we conducted post-hoc analyses. The results of these analyses are presented in Table 6 and Table 7. First, we probed the two-way interaction between EF and I_HLP on students’ final transcription scores (the intercept). A simple slopes analysis was conducted to examine the effect of EF at three levels of I_HLP.

As detailed in Table 6, the results revealed a conditional effect. The positive relationship between EF and baseline transcription was strong and statistically significant for children with high levels of independent home literacy practices (β = 0.60, *p* < .001) and for those with medium levels (β = 0.36, *p* = .006). However, for children with low levels of I_HLP, the effect of EF was not statistically significant (β = 0.12, *p* = .51). This pattern suggests that executive functions are most beneficial for children’s initial transcription abilities when they are also actively and independently engaged in literacy activities at home.

Next, we probed the three-way interaction between time, DF_HLP, and I_HLP on the growth of transcription skills. We examined the effect of I_HLP on the growth slope at different levels of DF_HLP (Table 7). Specifically, when formal practices at home were high, I_HLP acted as a significant accelerator of development. Pairwise comparisons confirmed that children with higher levels of independent practice showed significantly steeper and faster growth trajectories than those with lower levels (*p* = .01). In stark contrast, when formal practices were low, the benefits of independent practice vanished; the rate of growth was statistically indistinguishable across all levels of I_HLP (*p* = .95). This result is also reported in Figure 2.

### 3.6. Hierarchical Linear Modeling for Oral Language

We followed the same Hierarchical Linear Modeling procedure to model oral language trajectories. Intraclass coefficients again indicated significant clustering at the student (ICC = 0.35) and school (ICC = 0.11) levels, justifying a multilevel analysis. After evaluating several random effects specifications, the model with random intercepts and slopes at the student level was selected as the most parsimonious baseline model. The predictors and control variables were subsequently added to this model as fixed effects.

The final model included the predictors DI_HLP and DF_HLP, along with control variables SES, gender, and age, were added to the baseline model. The model specifying fixed effects on the intercept—but not on the slope—provided a reasonable fit, as indicated by model comparison (χ^2^(5) = 11.75, *p* = .038). This suggests that there is no significant interaction of home literacy practices on the slope of OL development.

Significant fixed effects on the intercept were observed for SES (β = 1.04, *p* = .048), indicating that children from higher socioeconomic backgrounds began the school year with stronger oral language skills. Dependent-informal home literacy practices showed a positive trend (β = 1.08, *p* = .076), although this effect did not reach conventional significance. DF_HLP, gender, and age did not significantly predict oral language at the start of the year.

A significant main effect of time was observed (β = 4.73, *p* < .001), confirming that oral language skills improved steadily throughout the kindergarten year (Table 8).

## 4. Discussion

The present study aimed to investigate the developmental trajectories of transcription and oral language skills in kindergarten students over the course of an academic year. Additionally, the study examined the influence of executive functions and home literacy practices on these literacy components. The results provide valuable insights into the interplay between cognitive and environmental factors in early writing development.

### 4.1. Developmental Trajectories of Transcription and Oral Language

The hierarchical linear modelling analyses revealed significant growth in transcription development over time, independent of the influence of external variables. However, the analyses revealed that both executive functions and independent home literacy practices significantly contributed to the transcription intercept positively. Specifically, higher executive function proficiency and greater independence in engaging with literacy activities at home were associated with higher baseline transcription scores at the beginning of the academic year. These findings align with previous research demonstrating the influence of executive function on transcription skills ([103]). Similarly, the role of independent home literacy practices is consistent with prior studies indicating their positive impact on transcription abilities ([7]; [54]; [94]). This study provides insight into how executive functions and children’s independence in engaging with home literacy activities contribute to transcription skills development in young children within a transparent orthographic system, along with their growth trajectory throughout the academic year. Furthermore, the significant interaction between executive function and independent home literacy practices offers a novel perspective on how they synergistically enhance transcription skills at the end of the course. However, dependent-formal home literacy practices did not present significant influence on transcription, contrasting with published articles ([1]; [6]; [7], [8]; [25]; [54]; [78]; [94]).

It is worth noting that previous studies have found a direct positive effect of dependent-formal home literacy practices on transcription skills ([1]; [6]; [7], [8]; [25]; [54]; [78]; [94]). However, most of these studies evaluated dependent-formal home literacy practices in isolation and did not analyze developmental trajectories over time. In our study, when executive functions, dependent-formal, and independent home literacy practices were included simultaneously, the isolated effect of dependent-formal practices on the transcription intercept was not significant. This suggests that dependent-formal practices alone may positively contribute to transcription if analyzed independently, but its effect is attenuated when accounting for other home literacy practices dimensions. In contrast, a significant three-way interaction was found among dependent–formal home literacy practices, independent home literacy practices, and the growth trajectory of transcription skills. This interaction indicates that dependent–formal home literacy practices substantially enhance the development of transcription skills when children also participate in independent literacy activities at home, highlighting the synergistic effect that arises when both types of home literacy practices are combined.

Children with low exposure to formal literacy practices at home do not exhibit differences in transcription skill growth over the academic year, regardless of their independence in literacy activities. However, as parents engage in more formal writing practices at home, children with greater independence benefit significantly more than those with less independence. Thus, the combined effect of these practices accelerated growth trajectory of transcription skills over the course of the academic year. This result highlights the importance of the home literacy practices carried out by both parent-guided and child-initiated home literacy practices in enhancing transcription development.

Regarding the results for oral language, only dependent-informal literacy practices showed a marginally significant influence on these skills, affecting the intercept scores but not the growth trajectory. Although the effect was not significant in our study, a substantial body of literature does indicate that playful literacy activities at home contribute to the development of oral language skills ([24]; [105]) but do not directly enhance writing skills ([1]; [19]; [54]; [70]).

Our findings indicate that executive functions significantly influenced transcription performance, whereas oral language skills did not exert a significant effect at this stage. It is worth noting, however, that this study focused on kindergarten children, a developmental stage in which transcription skills are still highly resource-demanding, whereas much of the empirical validation of these models has been conducted with elementary school students ([71]; [74]; [72]). This distinction highlights why executive functions may not yet exert a strong direct influence on oral language performance at this stage, as cognitive resources are primarily allocated to mastering transcription.

This finding aligns with the Not-So-Simple View of Writing model ([17]) and the Direct and Indirect Effects of Writing model ([73]), as well as with theoretical accounts of lower-level process automatization. According to these models, transcription skills demand substantial attentional and cognitive resources during early writing acquisition, leaving limited capacity to support other skills such as oral language production. As transcription becomes more automatized over time, cognitive resources are gradually freed, allowing executive functions to support higher-level writing processes, including idea generation, planning or revision.

Moreover, previous studies examining the influence of executive functions on oral language in kindergarten often assessed subcomponents of executive functions (e.g., attention, working memory, inhibition, cognitive flexibility) independently ([15]; [42]; [68]; [90]; [116]; [121]). In contrast, our study treated executive functions as a unified dimension, combining attention, working memory, inhibition, and cognitive flexibility. This methodological difference may contribute to the absence of a direct EF effect on oral language, as the influence of specific subcomponents could be diluted when aggregated into a single composite measure. Furthermore, it is important to consider how other foundational cognitive abilities, not measured in the present study, might work in combination with EF, as processing speed.

In addition, research conducted in transparent orthographies, such as Spanish, suggests that the influence of oral language on writing becomes more prominent in later academic years ([66]). This developmental trajectory aligns with theoretical propositions that transcription skills, initially resource-demanding, become automatized over time, thereby freeing cognitive resources for higher-level writing processes such as text generation and discourse planning ([14]; [17]; [73]). Previous research has shown that students face different challenges depending on the orthographic characteristics of their language ([10]; [32]; [89]), as shallow and opaque orthographies are processed and developed differently. This distinction may account for some of the differences observed in the findings. Therefore, it is crucial to expand the literature to include studies that examine writing development across both types of orthographic systems.

### 4.2. Limitations and Future Research

This study has several limitations. First, the sample size was constrained; although 365 students participated, only approximately one-third of families completed the home literacy questionnaire, reducing the final sample. Second, the sample was limited to kindergarteners in Santa Cruz de Tenerife, potentially limiting generalizability. Future research should expand sample size and include diverse Spanish regions. Third, the home literacy questionnaire, despite validated structure, had few items per home literacy practices type, possibly capturing incomplete data. Its focus on writing practices may have further constrained results. Future studies could develop more comprehensive questionnaires, incorporating items on reading, oral language, and other factors, and compare detailed versus concise versions to enhance construct validity. Third, it is important to note that executive functions and home literacy practices were assessed only at the end of the kindergarten year. This limits our ability to draw causal inferences regarding their influence on the initial level and growth of transcription and oral language skills. Future research should consider measuring these predictors at multiple time points throughout the school year to better capture potential reciprocal and bidirectional developmental processes. Despite this limitation, the current findings provide preliminary evidence on how executive functions and home literacy practices may relate to early literacy trajectories. Finally, our study focused on a specific set of predictors. Future research would benefit from incorporating other relevant cognitive abilities, such as processing speed, to build a more comprehensive model of early writing development and to better understand how these different skills work in combination with executive functions.

### 4.3. Educational Implications

This study’s findings yield key educational implications for early childhood literacy instruction, focusing on writing development. They highlight the interplay of cognitive (e.g., executive functions) and environmental factors (e.g., home literacy practices) in enhancing young children’s literacy outcomes. Higher executive functions skills, including working memory, cognitive flexibility, and inhibitory control, correlate with stronger transcription skills, suggesting that kindergarten programs should include activities like planning and task organization to boost executive functions and support writing development. Similarly, independent home literacy practices significantly enhance transcription skills. Educators should guide parents to foster independent reading and writing at home, perhaps by offering strategies for creating literacy-rich environments. Moreover, combining formal and independent home literacy practices accelerates transcription skill growth, emphasizing the need for structured instruction alongside open-ended, self-directed tasks. Additionally, dependent-informal home literacy practices bolster oral language skills, indicating that playful activities—such as drawing, letter sound games, or tracing—can strengthen oral language, a foundation for later writing development.

## 5. Conclusions

In conclusion, this study provides valuable insights into the developmental trajectories of transcription and oral language skills in kindergarten students, highlighting the critical role of executive functions and home literacy practices in shaping these skills over the academic year. The findings emphasize the significant contributions of executive functions and independent home literacy practices to the development of transcription skills, as well as the interaction between these factors and formal literacy practices at home, which accelerate transcription skill growth. However, home literacy practices and executive functions were not found direct effect on growth trajectories for oral language. Consistent with theoretical accounts of early writing ([17]; [73]), the present pattern, greater executive functions involvement in transcription than in oral language at kindergarten, provides a plausible interpretation of our results. Despite certain limitations, such as sample size and the regional scope of the study, these results underscore the importance of both cognitive and environmental factors in early literacy development, with implications for educational practice, particularly in fostering effective home literacy environments. Future research could expand upon these findings by addressing the limitations and exploring the role of various literacy-related skills in different age groups and orthographic contexts.

## Figures and Tables

**Figure 1 jintelligence-13-00163-f001:**
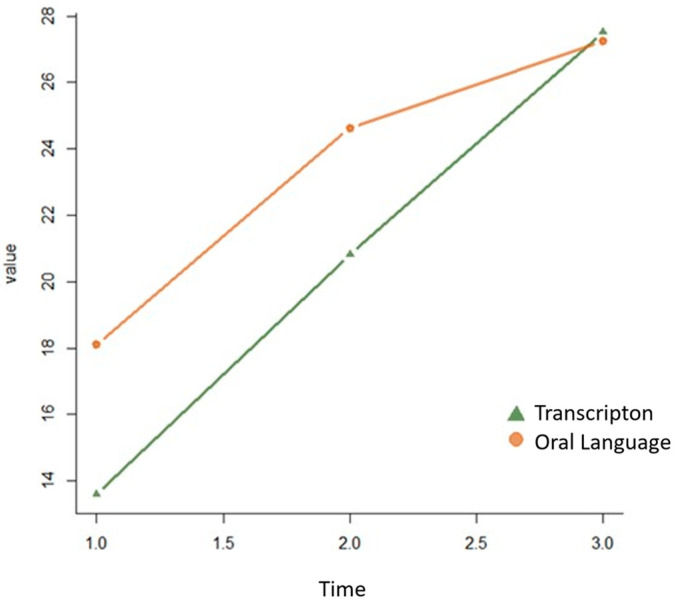
Growth curves for transcription and oral language measurements over time.

**Figure 2 jintelligence-13-00163-f002:**
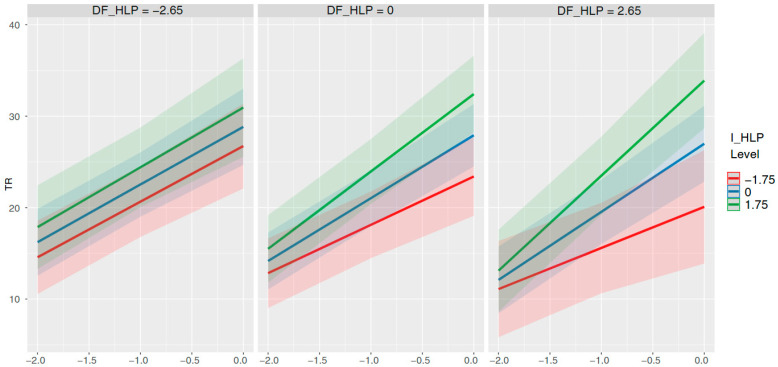
Hierarchical lineal modeling for transcription: significant effects on intercept and slope through the interaction between DF_HLP, I_HLP and time. Time: At the beginning (October; −2); in the middle (February; −1); and at the end of the course (May; 0 = intercept).

**Table 1 jintelligence-13-00163-t001:** Descriptive measures for longitudinal variables.

Time	LT	MS	n	Mean	sd	min	max	sk	kur
1	TR	CL	128	1.45	1.72	0	5	0.85	−0.71
		NW	128	7.63	2.58	0	10	−1.54	1.91
		PI	128	0.41	0.78	0	5	2.65	9.51
		EV	124	4.57	8.26	0	36	2.13	3.59
	OL	UW	130	8.19	10.92	0	43	1.35	1.17
		TU	130	8.58	15.21	0	75	2.50	6.57
		CS	130	2.12	2.87	0	11	1.42	1.53
	TR ^1^		124	14.02	9.51	0	61	1.64	2.68
	OL ^1^		130	18.89	28.15	0	129	1.98	3.95
2	TR	CL	130	2.71	1.74	0	5	−0.23	−1.33
		NW	130	7.93	2.17	0	10	−1.66	2.96
		PI	130	0.89	1.26	0	5	1.70	2.62
		EV	127	10.15	12.47	0	36	0.89	−0.75
	OL	UW	130	11.25	11.81	0	50	0.86	0.36
		TU	130	13.15	16.22	0	91	1.78	4.58
		CS	130	2.03	2.40	0	13	1.57	3.25
	TR ^1^		127	21.65	14.03	1	54	0.76	−0.72
	OL ^1^		130	26.43	29.99	0	152	1.36	2.49
3	TR	CL	130	3.11	1.78	0	5	−0.52	−1.16
		NW	130	8.74	1.82	1	10	−2.95	9.74
		PI	130	2.00	2.47	0	11	1.69	2.82
		EV	128	14.16	12.95	0	36	0.47	−1.34
	OL	UW	130	11.86	10.18	0	43	0.77	0.33
		TU	130	13.58	14.07	0	75	1.56	2.93
		CS	130	2.89	2.62	0	11	0.91	0.57
	TR ^1^		128	28.19	15.12	1	60	0.39	−1.16
	OL ^1^		130	28.32	26.50	0	129	1.16	1.45

*Notes*: LT = latent variable: MS = measures; TR = Transcription; OL = Oral Language; CL = Copying Letters; NW = Names Writing; PI = Phoneme Isolate; EV = Expressive Vocabulary; UW = Unique Words; TU = T-Unit; CS = Correct Sequences; sd = standard deviation; min = minimum value; max = maximum value; sk = skewness; kur = kurtosis; ^1^ = variable composed of its measures.

**Table 2 jintelligence-13-00163-t002:** Descriptive measures for non-longitudinal variables.

LT	MS	n	Mean	sd	Min	max	sk	kur
EF	AT	130	16.92	6.44	4	32	0.49	−0.37
	IC	130	0.66	0.21	0	1.16	−0.66	1.33
	CF	130	0.20	0.10	0.06	0.78	2.34	8.67
	DB	130	2.22	1.03	0	4	−0.86	0.37
	OC	130	1.42	1.42	0	7	0.95	1.08
DF_HLP	DF_1	130	2.86	1.00	1	4	−0.71	−0.52
	DF_2	130	2.18	1.24	1	4	0.30	−1.60
	DF_3	130	2.74	0.94	1	4	−0.57	−0.55
DI_HLP	DI_1	130	3.01	0.70	1	4	−0.55	0.63
	DI_2	130	2.82	0.92	1	4	−0.49	−0.54
	DI_3	130	2.53	0.86	1	4	−0.25	−0.64
	DI_4	130	2.46	0.90	1	4	−0.13	−0.82
	DI_5	130	2.25	1.04	1	4	0.06	−1.34
I_HLP	I_1	130	3.42	0.75	1	4	−1.28	1.33
	I_2	130	3.43	0.78	1	4	−1.41	1.62
	I_3	130	3.61	0.69	1	4	−1.89	3.42
EF ^1^		130	21.43	7.48	4.95	41.06	0.26	−0.22
DF_HLP ^1^		130	7.77	2.64	3	12	−0.08	−0.97
DI_HLP ^1^		130	13.07	3.26	5	20	−0.14	−0.28
I_HLP ^1^		130	10.46	1.75	3	12	−1.29	1.82

*Notes*: AT = attention task; IC = Inhibitory Control; CF = Cognitive Flexibility; DB = Digit-span Backward; OC = Oral Cloze task; EF = executive function, composed variable of its measures; DF_HLP = dependent-formal home literacy practices; DI_HLP = dependent-informal home literacy practices; I_HLP = independent home literacy practices; sd = standard deviation; min = minimum value; max = maximum value; sk = skewness; kur = kurtosis; ^1^ = variable composed of its measures.

**Table 3 jintelligence-13-00163-t003:** Centrality measures of network analysis.

Vertex	Degree	Eigen	Closeness	Bet	Group
DF_1	12	0.90	0.07	0.20	1
DF_2	10	0.77	0.07	0.00	1
DF_3	14	1.00	0.09	1.43	1
DI_1	14	0.94	0.09	4.65	1
DI_2	14	1.00	0.09	1.43	1
DI_4	14	1.00	0.09	1.43	1
DI_5	12	0.89	0.08	0.65	1
I_1	4	0.15	0.05	0.00	2
I_2	12	0.69	0.08	7.55	2
I_3	6	0.28	0.06	1.65	2

*Notes*: DF = items for dependent-formal home literacy practices; DI = items for dependent-informal home literacy practices; I = items for independent home literacy practices; bet = betweeness.

**Table 4 jintelligence-13-00163-t004:** Correlation between latent variables.

	DF_HLP	DI_HLP	I_HLP	EF
DF_HLP				
DI_HLP	0.55 ***			
I_HLP	0.15 *	0.30 ***		
EF	−0.34 ***	−0.09	0.08	
TR_1	−0.28 **	−0.05	0.09	0.15
TR_2	−0.16 *	0.02	0.25 **	0.30 ***
TR_3	−0.14	−0.02	0.22 *	0.41 ***
OL_1	0.09	0.08	0.04	0.06
OL_2	0.11	0.18 *	0.04	0.04
OL_3	0.21 *	0.20 *	0.10	0.02

*Notes*: TR = Transcription latent variable in moment 1 (TR_1), moment 2 (TR_2) and moment 3 (TR_3); OL = Oral Language latent variable in moment 1 (OL_1), moment 2 (OL_2) and moment 3 (OL_3); DF_HLP = dependent-formal home literacy practices; DI_HLP = dependent-informal home literacy practices; I_HLP = independent home literacy practices; EF = executive function; * = *p* < .05; ** = *p* < .01; *** = *p* < .001.

**Table 5 jintelligence-13-00163-t005:** Linear mixed model fit for transcription: fixed effects and analysis of deviance.

	Analysis of Deviance			Fit		
Predictor	Chisq	Df	Pr (>Chisq)	Estimate	Std. Error	t Value
(Intercept)	202.98	1	<0.000 ***	27.09	1.90	14.25
Time	119.04	1	<0.000 ***	6.80	0.62	10.91
DF_HLP	0.62	1	0.43	−0.36	0.46	−0.79
I_HLP	11.94	1	<0.000 ***	2.53	0.73	3.46
EF	18.84	1	<0.000 ***	0.72	0.17	4.34
SES	5.34	1	0.01 *	0.52	0.23	2.31
Gender	0.71	1	0.38	1.43	1.70	0.84
Age	0.19	1	0.66	1.29	2.92	0.44
Time * DF_HLP	0.79	1	0.37	0.21	0.23	0.89
Time * I_HLP	5.84	1	0.01 *	0.89	0.37	2.42
DF_HLP * I_HLP	4.15	1	0.04 *	0.51	0.25	2.04
Time * EF	19.47	1	<0.000 ***	0.36	0.08	4.41
DF_HLP * EF	0.53	1	0.47	0.05	0.06	0.73
I_HLP * EF	5.37	1	0.02 *	0.22	0.10	2.32
Time * DF_HLP * I_HLP	5.12	1	0.02 *	0.29	0.13	2.26
Time * DF_HLP * EF	1.09	1	0.29	0.03	0.03	1.05
Time * I_HLP * EF	3.24	1	0.07	0.09	0.05	1.80
DF_HLP * I_HLP * EF	0.21	1	0.65	0.02	0.04	0.46
Time * DF_HLP * I_HLP * EF	0.62	1	0.43	0.01	0.02	0.79

*Notes*: DF_HLP = dependent-formal home literacy practices; I_HLP = independent home literacy practices; EF = executive functions; SES = socioeconomic status; * = *p* < .05 ; *** = *p* < .001.

**Table 6 jintelligence-13-00163-t006:** Simple comparison of the effect of EF on transcription at the final assessment across levels of I_HLP.

Levels of I_HLP	Effect of EF	Std. Error	df	t Value	*p* Value
−1.75	0.12	0.19	121	0.66	.51
0	0.36	0.13	119	2.82	.006 **
1.75	0.60	0.18	119	3.40	<.001 ***

*Notes*: I_HLP = Independent home literacy practices; EF = Executive functions; ** = *p* < .01; *** = *p* < .001.

**Table 7 jintelligence-13-00163-t007:** Simple slope comparisons of time at different levels of I_HLP within each DF_HLP condition.

Levels of DF_HLP	Effect of I_HLP	Estimate	Std. Error	df	t Value	*p* Value
−2.65	−1.75	−0.23	0.72	119	−0.32	.95
	0	−0.46	1.43	119	−0.32	.95
	1.75	−0.23	0.72	119	−0.32	.95
0	−1.75	−1.59	0.65	126	−2.45	.04 *
	0	−3.18	1.30	126	−2.45	.04 *
	1.75	−1.59	0.65	126	−2.45	.04 *
2.65	−1.75	−2.95	1.02	129	−2.89	.01 **
	0	−5.90	2.04	129	−2.89	.01 **
	1.75	−2.95	1.02	129	−2.89	.01 **

*Notes*: DF_HLP = Dependent-formal home literacy practices; I_HLP = Independent home literacy practices; * = *p* < .05; ** = *p* < .01.

**Table 8 jintelligence-13-00163-t008:** Linear mixed model fit for oral language: fixed effects and analysis of deviance.

	Analysis of Deviance			Fit		
Predictor	Chisq	Df	Pr (>Chisq)	Estimate	Std. Error	t Value
(Intercept)	105.17	1	<0.001 ***	31.17	3.04	10.26
Time	12.72	1	<0.001 ***	4.73	1.33	3.57
DF_HLP	0.04	1	0.84	0.17	0.88	0.20
DI_HLP	3.16	1	0.07	1.08	0.61	1.78
SES	3.92	1	0.04 *	1.04	0.53	1.98
Gender	0.99	1	0.31	−3.98	3.98	−0.99
Age	1.78	1	0.18	8.83	6.62	1.33

*Notes*: DI_HLP = dependent-informal home literacy practices; DF_HLP = dependent-formal home literacy pracices; SES = socio-economic status level; * = *p* < .05; *** = *p* < .001.

## Data Availability

The data are not publicly available due to ongoing research by the project team. Anonymized data may be made available upon reasonable request after the completion of the current studies.

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
