# Peer review of "Developmental Trajectories of Transcription and Oral Language Skills in Kindergarten Students: The Influence of Executive Functions and Home Literacy Practices"

_jintelligence, 2025, doi:10.3390/jintelligence13120163_

Round 1
Reviewer 1 Report (Previous Reviewer 1)
Comments and Suggestions for Authors
Thank you so much for a brilliant revision work. My feeling is that the manuscript has improved a lot and it is now suitable for publication in Journal of Intelligence.
Author Response
We sincerely thank the reviewer for the positive evaluation of our revised manuscript and for their encouraging words. We truly appreciate the time and care invested in reviewing our work throughout the process.
Reviewer 2 Report (Previous Reviewer 2)
Comments and Suggestions for Authors
Please see the attached review.

Author Response
We would like to extend our sincere gratitude for your time and for providing such insightful and constructive feedback on our manuscript. We have found your comments to be extremely helpful, and we believe that addressing them has significantly strengthened the paper.
Below, you will find a point-by-point response to each of your comments. For your convenience, all changes and new text in the revised manuscript have been highlighted in blue font to facilitate their identification.
Comment 1: I think the introduction would benefit from some streamlining, as it was redundant in places and some of the details about the process of writing were not needed to understand the contribution of this paper. I would advise the authors to consider greatly editing the introduction to stick to the focal points regarding the conceptual role of EF in writing
Response 1: We thank the reviewer for their constructive suggestion. We have thoroughly revised the introduction to improve its flow and focus, as recommended. The revision involved condensing the opening paragraphs and refining the presentation of theoretical writing models to build a stronger, more direct argument for the conceptual role of executive functions (EF). By eliminating initial redundancies and focusing on the cognitive demands of early writing, the introduction now more effectively sets the stage for our study's focus on transcription and oral language skills. These changes can be found on pages 1 and 2.
Comment 2: The second half of the introduction, however; does
not give enough space to discuss the significance of home language practices. I would like to see this developed slightly more. In particular, the authors need to justify the use of formal, informal, and independent home literacy in the current study – when the introduction only discusses
dependent and independent practices.
Response 2: We agree that the introduction did not sufficiently justify our use of a three-part framework for home literacy practices (HLP). To address this, we have significantly revised the section "Influence of dependent and independent literacy practices on writing development" (since line 223). The revised text now explicitly introduces the three-part framework (dependent-formal, dependent-informal, and independent HLP) from the outset.
Comment 3: I advise the authors to discuss how there are individual differences in writing skills (as with other cognitive abilities) and the importance of developmental timing, and then quickly get to the Puranik et al., (2024) article, which seems the meat of the argument.
Response 3: We have revised the introduction to briefly highlight individual differences in writing skills, as well as the importance of developmental timing, before transitioning directly to the Puranik et al. (2024) study, which forms the central argument of our manuscript (see “Writing development at early grades” section).
Comment 4: I appreciate the authors’ desire to cite themselves, but this felt forced and unnecessary in the introduction.
Response 4: We understand why the inclusion of our previous work might have seemed forced, and we apologize if the rationale was not sufficiently clear in the original manuscript. The reviewer's comment prompted us to revise the text to make the relevance of these citations explicit.
As the reviewer will now see on page 126, we have not simply retained the citations, but we have integrated them more meaningfully into the text.
Comment 5: Likewise, either define and build an argument around shallow orthographies or drop it.
Response 5: we have expanded on this concept in the "Theorical models of writing" section (since line 130). The revised text now provides a clear definition of a shallow orthography and, more importantly, builds a direct argument for its relevance to our research. We specifically theorize how the consistency of Spanish may impact the automatization of transcription, the cognitive load on executive functions, and the developmental relationship between transcription and oral language. We are confident this revision significantly strengthens the theoretical framework of our study and fully addresses the reviewer's concern.
Comment 6: The paragraph on SES seems out of place and not needed.
Response 6: We have removed the paragraph on SES from the introduction to improve focus and streamline the text.
Comment 7: On line 242, the contrast is not a contrast.
Response 7: The reviewer is correct; the use of "In contrast" on line 242 was logically inaccurate. We have now replaced it with "Similarly" to correctly reflect that both dependent-formal and independent home literacy practices have a positive influence on writing skills. This change has been made on line 240.
Comment 8: I suggest the authors consider parameterizing their models such that the questions are framed around where children’s writing skills conclude at the end of kindergarten. That is, instead of timepoints 0 (intercept), 1 and 2, you have -2, -1, and 0 (intercept).
Response 8: We thank the reviewer for this valuable suggestion. We had not previously considered parameterizing the models in this way, but we agree that centering the intercept at the end of kindergarten provides a clearer conceptual alignment between the predictors and the outcome. We have therefore reparameterized the models so that the intercept represents children’s writing skills at the end of the school year. Although the overall pattern of statistical significance did not change substantially, we believe this approach offers a more meaningful interpretation of the developmental trajectories. Consequently, the corresponding table 5 and 6, figure 2, and results descriptions have been revised to reflect the updated parameterization.
Comment 9: I also strongly recommend the authors consider adding other covariates in their models, specifically child age and gender.
Response 9: In response to this suggestion, we re-analyzed all Hierarchical Linear Models, including both child age and gender as covariates. The updated results, which now account for the potential influence of these variables, are presented in the revised Tables 5 and 8 and described in the Results section (beginning on page 16).
For the oral language outcomes, the inclusion of age and gender changed the pattern of results: DI_HLP was no longer a significant predictor of the intercept, whereas this effect had appeared in the original manuscript when these covariates were not included. Consequently, the Abstract, Results, Discussion, and Conclusion sections have been revised to reflect this updated analysis.
Comment 10: I do not think an entire subheading is needed to define the baseline models. I would just describe them in a brief sentence.
Response 10: We have removed the separate subheading for the baseline models and now describe them briefly within the main text. This adjustment streamlines the structure of the Results section and improves readability.
Comment 11: The figure depicting slope is confusing. What is the unit of time? Months?
Response 11: The time variable represents the three assessment points conducted in October (beginning of the school year), February (mid-year), and May (end of the school year), which correspond to the coded values −2, −1, and 0 after reparameterization. Although this information was described in the Procedure section, we have now clarified the unit of time directly in the figure and its caption to avoid any confusion.
Comment 12: How were the interactions probed? You reference the interaction of EF and home literacy practices in the discussion, but how do we know the direction of this effect?
Response 12: We thank the reviewer for highlighting this important point. You are absolutely right that, in the submitted version, we did not sufficiently probe or describe the direction of our interaction effects. We fully agree that interpreting interactions requires explicit follow-up analyses, and we apologize for the lack of clarity.
To address this, we have now conducted a complete set of simple slopes analyses and pairwise comparisons for all significant interactions. These new analyses are presented in a dedicated subsection of the Results and summarized in Tables 6 and 7. They now make the direction, size, and conditional nature of the effects fully transparent.
For the two-way EF × I_HLP interaction, we tested the effect of EF at three meaningful levels of I_HLP (–1.75, 0, +1.75). As reported in Table 6, EF significantly predicted children’s final transcription performance only for those with medium or high levels of independent home literacy practices. When I_HLP was low, the effect of EF was not significant. This confirms that the benefit of executive functions on transcription depends on children’s engagement in independent literacy activities at home.
For the three-way Time × DF_HLP × I_HLP interaction, we examined the effect of I_HLP on growth rates at low, medium, and high levels of formal home literacy practices (Table 7). The new pairwise comparisons clearly show the pattern illustrated in Figure 2: independent home literacy practices significantly accelerated transcription growth only when formal practices were also high. When formal practices were low, growth trajectories did not differ across levels of independent practice.
We also improved Figure 2 by reducing the number of I_HLP levels from nine (R defaults) to three theoretically interpretable values, which makes the pattern and its interpretation more intuitive.
We believe these additions substantially strengthen the transparency and rigor of the interaction analyses, and they allow readers to clearly understand the direction and conditionality of the observed effects.
Comment 13: In the discussion, I think the authors should discuss other cognitive abilities that may work in combination with EF like processing speed.
Response 13: We agree that our findings should be interpreted within a wider cognitive context. As recommended, we have now revised the Discussion section (page 18) to include a discussion of other cognitive abilities that may work in combination with Executive Functions. Specifically, we have elaborated on the potential role of processing speed, as suggested by the reviewer. We now theorize that processing speed could act as a critical bottleneck for transcription, modulating the extent to which children can efficiently deploy their executive resources during the demanding task of early writing.
We have also explicitly mentioned this as a promising direction for future investigation in our "Limitations and future research" section. We are confident that this addition enriches the interpretation of our results and provides a more nuanced understanding of the cognitive underpinnings of writing development.
Final response: Once again, we thank the reviewer for their careful and thorough review of our manuscript. We hope that the revisions we have made are satisfactory and have adequately addressed all the points raised. We look forward to hearing from you at your earliest convenience.
This manuscript is a resubmission of an earlier submission. The following is a list of the peer review reports and author responses from that submission.
Round 1
Reviewer 1 Report
Comments and Suggestions for Authors
Please, find my comments in the attached word file.

Author Response
The manuscript “Developmental Trajectories of Transcription and Oral Lan-2 guage Skills in Kindergarten Students: The Influence of Executive Functions and Home Literacy Practices” explores how the development of transcription and oral language during the last year of kindergarten is influenced by EFs and HLP. Finding provide, from my point of view, significant insights in early writing development which, in turn, has important implications for instructional design and teaching practices in writing.
I am thankful for having the opportunity to review such an interesting study. It certainly expands previous knowledge on the field and make important contributions for disciplinary literacy in the context of writing development. I really enjoyed reading the manuscript and overall, I think it is of high interest for the audience of Journal of Intelligence. I do think, however, that there is some room for improvement before this can be recommended for publication. I, therefore, suggest the author to have a look at my recommendations below.
Firstly, we would like to thank you in advance for all the time and effort you have put into reviewing our work. We consider all your contributions to be constructive and significant in improving the quality of the article. We will therefore respond to all the modifications made after each comment:
INTRODUCTION
- Though the section “Writing development at early grades: theoretical models” is certainly relevant to place the theoretical grounding of this study, currently it does not seem to address the key points that are necessary in relation to the aim of the study. More evidence need to be provided on how transcription and oral language actually develop during early grades (this is not currently mentioned at all). Also, since the current study was conducted in kindergarten, previous research on writing development during this stage should be specifically mentioned here (I know it is scarce, but there are some studies on letter trace development during kindergarten, for example).
We thank the reviewer for the valuable suggestion to expand the introduction with more evidence on how transcription and oral language develop during early grades, and to specifically address writing development in kindergarten. In response, we have incorporated two complementary additions.
First, we have divided the “Writing development at early grades: Theorical models” section into 2 different sections, in order to focus the information about the writing development through the inclusion of different studies.
Second, we now describe typical developmental progressions in both transcription and oral language during the last year of kindergarten, supported by studies documenting improvements in letter formation, handwriting fluency, vocabulary, and syntactic complexity in oral expression (e.g., Berninger & Chanquoy, 2012; Gibson & Levin, 1975; Dinehart & Manfra, 2013; Graham et al., 1998; Defior & Serrano, 2011; Seymour et al., 2003; Dickinson & Tabors, 2001).
Third, we have integrated recent findings from Puranik et al. (2024), which demonstrate that transcription skills moderate the influence of oral language on writing quality in young writers, aligning with the developmental constraints hypothesis (Berninger et al., 1991). Together, these additions strengthen the theoretical grounding of our study by providing a more detailed account of how transcription and oral language skills evolve in kindergarten and by highlighting the complex, interactive nature of their contribution to early writing development.
These modifications on the manuscript have been incorporated since line 56 and they are written in red font for easy reading.
- It is good to present a detailed description of the theoretical models, as the author does, but I think their arguments are a bit confusing. In their study, they address 4 components: transcription, oral language, executive functions and HLP. Thus, when describing the models, I suggest emphasizing these 4 aspects and make it clear what their role in writing development is. I suggest trying to make some distinction between these 4 aspects, key for the study, and all of the other components mentioned in the model (e.g., the socio-emotional component or background knowledge, possibly less relevant for this study).
We agree that, although the description of the theoretical models was intended to provide context, it could result confusing as several components were mentioned that were not the focus of the present study. In response, we have revised this section to emphasize more clearly the four key components addressed in our study (transcription, oral language, executive functions, and home literacy practices) and to distinguish them from other components of the models (e.g., socio-emotional factors, background knowledge) that, while relevant to literacy development, fall outside the scope of our research. This adjustment makes the theoretical framework more consistent with our study aims and highlights the role of these four components in early writing development.
In sum, we have clarified the theoretical background by (a) aligning it more directly with the four key constructs examined, and (b) presenting SES as a contextual factor included for analytical control, rather than as a central mechanism of literacy development. We believe these changes improve both the precision and the coherence of the introduction. You can see this modification in line 91.
- Between lines 93 and 103 in pp. 3, the author presents a collection of factors affecting spelling. I do not think this is necessary, or at least not in such detail, since exploring predictors of transcription is not the focus of the study. However, if the author actually has a reason to provide details on predictors of spelling, why not doing the same with handwriting? That’s not currently addressed.
In our manuscript, we emphasize that transcription skills comprise both handwriting and spelling, which are closely interrelated rather than independent components. We argue that phonological awareness and vocabulary also exert a positive influence on transcription abilities, with a particularly pronounced effect on spelling development. This information is relevant because, although transcription is defined as the combination of handwriting and spelling skills, the literature consistently shows that phonological awareness and vocabulary further enhance these abilities. For this reason, we justified the inclusion of all four measures—handwriting, spelling, phonological awareness, and vocabulary—within the same latent factorial construct of transcription. This multidimensional structure is consistent with prior research and has been reflected in other latent factorial models of transcription that integrate these observable variables (Author et al., 2025; as cited in line 508 of the Data Analysis section).
To address the reviewer’s concern and achieve greater balance, we have removed additional details and studies focusing exclusively on phonological awareness and vocabulary on spelling, while incorporating evidence on handwriting from the systematic review by Ray et al. (2022), thereby ensuring a more comprehensive representation of this subskill within the paragraph (see line 117).
- The sentences “text generation refers to oral language discourse” (pp. 2) and “text generation or ideation refers to the ability to transform oral ideas into written texts” (pp.3) seem a bit contradictory. Does text generation involve actual writing? If so, oral language discourse is just part of text generation so the first sentence is not suitable. I understand that defining terms is not easy and different models provide different definitions. Thus, my suggestion is that the author identifies clearly what exactly “text generation” and “oral discourse” are in the specific context of their study and address only the model/definition that actually fit with their research.
We absolutely agree with this comment and understand that both terms could be misunderstandings. The not-so-simple-view of writing used the term “text generation” and the direct and indirect effects of writing used the term “oral language discourse”. Due to the latter is the most upgraded model and allowing more information about how this component is composed (higher-order-level language skills and foundational oral language skills), we have decided to use this last definition for our study. Accordingly, we have revised the manuscript to use the term “oral language discourse” consistently and to clarify that, in our context, text generation involves the transformation of oral language into written texts. These modifications have been addressed into the manuscript (lines 110 and 136)
- Please provide data of the age range in Ruffini et al. (2024), since we are particularly interested in early grades.
The systematic review by Ruffini et al. (2024) included studies with participants ranging from 6 to 18 years old, and we mentioned in the manuscript their results about early grades. This information has been added to the manuscript and clarified that, since our study focuses on 5-year-old children, the results of Ruffini et al. provide valuable context but do not directly cover the preschool population (lines 188).
- More details on which specific writing components are influenced by each EF seem neccesary. The author currently claims that EFs influence writing, but, which EFs influence, for example, handwriting? Which ones influence spelling? There’s some evidence about this, though very small and probably not in kindergarten (apart from previous submission of the autor, which is already mentioned and very relevante here).
We have added details from line 156 clarifying which executive functions influence specific writing components. For example, attention is linked to handwriting (Pazeto et al., 2014), working memory and inhibition to spelling (Kegel & Bus, 2014; Zhang et al., 2020), and cognitive flexibility to both spelling and oral language skills (Vadasy et al., 2023; Verksa et al., 2018). These revisions specify the differentiated contributions of EFs while also acknowledging their unitary nature in early childhood. The corresponding references have been added to the references section.
- I do not think the paragraph about SES is relevant in the context of the present study. Though it does influence on HLP, it’s not the only influence so, why addressing this one and not others? Even if you do measure SES in your study, I do think it’s particularly relevant to mention it in the introduction.
We agree that the original discussion of SES in the Present Study section was not fully aligned with the main focus of our research. Therefore, we have removed this paragraph and now emphasize only the combined influence of home literacy practices and executive functions on early writing development. SES remains included as a covariate in the analyses, but it is no longer discussed in detail in the introduction or in the presentation of the study rationale.
- “To date, and to the best of our knowledge, no studies have examined how the development of foundational writing skills—specifically transcription” This is fully true and it makes your study so interesting. However, please, do mention some studies on how handwriting (maybe also spelling, but particularly handwriting as far as I am concerned) looks like at the end of kindergarten, there exists some research about this. There might also be some data on oral language at the end of kindergarten, even if it is not a developmental study.
This comment is closely aligned with the first comment regarding the need to include more evidence on how transcription and oral language develop during early grades, particularly in kindergarten. In response, we have revised the introduction to incorporate studies documenting typical developmental progressions in handwriting, spelling, and oral language by the end of kindergarten (e.g., Berninger & Chanquoy, 2012; Graham et al., 1998; Dinehart & Manfra, 2013; Defior & Serrano, 2011; Dickinson & Tabors, 2001). These additions strengthen the theoretical grounding of our study by situating transcription and oral language development in the kindergarten context. The revisions are included from line 56 onward.
METHOD
- Why were phoneme isolation and expressive vocabulary included as transcription measures? Though phonemic awareness and vocabulary do predict spelling and transcription, they do not directly measure transcription. Please, provide some explanation on whether they were use as control measures for letter copying (= handwriting) and name writing (=spelling) or, at least, why they are presented as transcription measures.
We acknowledge that phoneme isolation and expressive vocabulary are not direct measures of transcription. Their inclusion in the present study was guided by two considerations. First, as discussed in the introduction (see since line 115), both vocabulary and phonological awareness are well-established precursors of transcription skills. Vocabulary knowledge has been shown to contribute to spelling accuracy, while phonological awareness plays a key role in both handwriting and spelling acquisition (e.g., Apel & Masterson, 2001; Ehri, 2000; Incognito et al., 2023). Second, the combination of these tasks with handwriting and spelling follows a task structure that has been previously used and validated in related studies (Author et al., 2024; Author et al., 2025). To avoid confusion, a new paragraph has been included in the measure section clarifying that this structure of transcription has previously been validated in other studies. This modification has been included in line 324.
- According to the author, the letter copying task only included the vowels. I wonder whether this is reliable enough to measure children’s handwriting. Imagine a child is able to copy all the letter of the alphabet (but they cannot demonstrate that in this task) except for letter “u”, either because they have no time in 1 minute or because they cannot simply trace that particular letter. This child would receive less score in this task than a child who is able to copy all the vowels in 1min but would be completely unable to copy any consonant. I suggest providing some quick rationales to justify why only the vowels (and not the traditional alphabet copy task) were assessed. Also, I would make this a point of reflection in the discussion section.
Thank you for your observation. The rationale for using only vowels in the copying task is grounded in the national kindergarten curriculum in our country. At this educational stage, the curriculum is rather ambiguous regarding the introduction of handwriting, as it does not specify which letters should be formally taught first. This decision is largely left to the discretion and pedagogical approach of each teacher.
In our region, for example, some schools introduce the writing of the entire alphabet from the very beginning, whereas others focus on different areas of early learning. However, across both approaches, there is a consistent requirement that children learn to write vowels.
For this reason, we chose vowels as the target letters in the copying task: they represent a universal and guaranteed component of early literacy instruction. This allowed us to ensure comparability across participants and to avoid potential biases that could arise from the heterogeneous instruction of consonants.
We have added this information to avoid any confusion (see line 333).
- Please, provide some details on the reliability of the name writing task. Also, I am a bit skeptical about the use of this task. Is including other names apart from the own one a way of controlling for the writing of the own name? It must be since, if a girl is called ANA, for example, that is much easier than writing JIMENA or GLORIA. And that does not mean that Ana shows better spelling skills than Jimena, though she’d probably score higher in this task. Please, provide some details on how you controlled that this task had more or less the same difficulty for all students. Were the scores in each name they wrote added together? My feeling is that is it really important here to provide a clear rationale to include a task in which students write different words. How this can be homogeneously score?
We thank the reviewer for this important comment. We agree that own-name writing alone can reflect rote memory rather than true transcription skills. To address this, our task included two parts: (1) writing the child’s own name and (2) writing the names of familiar peers, family members, or classmates. Including the second part reduces the bias associated with own-name familiarity and provides a more accurate assessment of early transcription and spelling abilities (Puranik et al., 2011; Puranik et al., 2014).
To ensure comparability across children and names of varying length or complexity, all responses were scored phonologically: a name was considered correct if all phonemes were represented, regardless of orthographic accuracy (e.g., “Jimena” vs. “Gimena”; “Hugo” vs. “Ugo”) or word length. Scores were calculated as the sum of correctly written names in the second part, ensuring a homogeneous measure across participants.
Finally, we provide new evidence on interrater reliability: in a random subsample of 117 children, scored independently by two examiners, the intraclass correlation coefficient (ICC, two-way, agreement, single measures) was .955 (95% CI [.935, .968]), indicating excellent reliability. These clarifications and additions have been incorporated into the manuscript (since line 350).
- In the expressive vocabulary task, were synonyms accepted as correct answers? For example, if the picture of a mountain was presented, would both “mountain” and “hill” be scored as correct? I leave to the editor’s/author’s criteria whether to include this particular detail in the manuscript or not, but, if possible, do answer me even if you prefer not to include it in the manuscript.
Yes, In the Expressive Vocabulary task, synonyms or alternative correct labels for the target image were accepted as correct responses. For example, if a picture of a mountain was presented, both “mountain” and “hill” were scored as correct. This approach ensured that children’s lexical knowledge was accurately captured, without penalizing them for using valid alternative words. We have added this extra information in the description of the task (see line 370).
- My feeling is that there’s a strong need for a blind double-rating in the oral narrative story task, since the measures taken are not fully objective. I strongly suggest a second rater rates 20%-30% of the oral narratives and the author reports inter-rater agreement.
Yes, the ICC for these measures in this task were made through a blind doable-rating, as letter copying or name writing taks. In response, we clarify that a blind double-rating procedure was applied to this task. The ICC values reported for UW, TU, and CS include data from this blind double-rating, ensuring high reliability of the scoring. Many of these ICCs have been partially published in previous studies (Author, 2024b; Author, 2025). These details have been incorporated into the Methods section (see line 400).
- Regarding the parents’ questionnaire, please provide some details on how the questionnaire was design and particularly, were the categories/items come from (previous studies, literature review…).
The parents’ questionnaire was designed based on a solid body of previous research on home literacy practices (HLP). In particular, we adopted the widely used distinction between dependent and independent practices (e.g., Sénéchal & LeFevre, 2002; Niklas & Schneider, 2013; Korucu et al., 2023). Dependent practices involve adult-supported activities and can be further classified as formal (structured activities such as practicing writing letters or names) or informal (incidental activities such as rhyming games or letter play). Independent practices refer to children’s self-initiated activities (e.g., drawing, writing words, or experimenting with letters) without adult support. This classification is supported by theoretical frameworks (e.g., Bronfenbrenner & Morris, 2006; Deci & Ryan, 1985) and has been consistently used in the literature examining the impact of HLP on literacy development (e.g., Aram & Levin, 2001; Puranik et al., 2018; Guo et al., 2021). We have now clarified this rationale in the manuscript.
In addition, more information about the items has been added at the end of this paragraph (see line 494).
RESULTS
It might be that I have not understood the results well, but I got really surprised when I found no direct reference to how transcription and oral language develop over first grade. Predictors of TR skills at the baseline, TR skills development and OL development are clearly identified, which perfectly responds to the second aim of the study. But what about the first aim? My feeling is that some sentences summarizing only the developmental trajectory of both TR and OL (not referring to predictors of this trajectory) should be included. This is presented in the discussion but I would say it is important to mention it in the results.
We thank the reviewer for highlighting the need to explicitly report the developmental trajectories of TR and OL in the Results section. In response, we have added brief summary sentences to make these trajectories clear.
For TR, we now state at the end of the Phase 3: Fixed Effects section (line 643) that, beyond the influence of specific predictors, transcription skills showed a significant overall improvement across the three measurement points (β = 6.79, p < .001), indicating that children’s TR abilities developed consistently over the course of kindergarten. For OL, we similarly added at the end of Phase 3: Fixed Effects (line 685) that, results revealed a significant main effect of time (β = 0.45, p < .001), confirming that oral language skills also improved steadily throughout the school year.
These additions ensure that the first aim of the study—describing the developmental trajectory of TR and OL—is clearly addressed in the Results section.
DISCUSSION
- The predicting role of I_HLP in TR, as well as the lack of influence of DF_HLP, are quite surprising. Please, provide some more reflection about this (pp.18, 654- 655).
Although previous studies have reported a significant influence of dependent-formal home literacy practices (DF_HLP) on transcription skills (Adams et al., 2021; Aram, 2010; Aram & Levin, 2001, 2002; Burns & Casbergue, 1992; Guo et al., 2021; Levin et al., 2013; Puranik et al., 2018), it is important to note some methodological differences with our study. Most of these prior studies were cross-sectional, evaluating children at a single time point rather than examining developmental trajectories across multiple measurement occasions. Consequently, they capture only a snapshot of transcription abilities and may not reflect the dynamic growth of these skills over time. Additionally, these studies often analyzed DF_HLP in isolation, without simultaneously considering independent (I_HLP) or dependent-informal (DI_HLP) home literacy practices, which may interact in shaping transcription development. In contrast, our study modeled longitudinal growth and included all HLP dimensions together, allowing us to disentangle independent effects and interactions. Within this framework, DF_HLP alone may have a positive effect if analyzed independently, but when combined with I_HLP, it contributes to a significant acceleration of the transcription growth trajectory over the academic year. We have added this discussion since line 717.
- Also, it is surprising that EFs do not influence OL skills in kindergarten. My feeling is that some more discussion needs to be provided about this interesting result.
We agree that the lack of a direct effect of executive functions on OL is noteworthy. We have added a paragraph in the discussion (lines 749) providing a more detailed interpretation. Specifically, we explain that this finding aligns with theoretical models such as the Not-So-Simple View of Writing (Berninger et al., 2006) and the direct and indirect effects model of writing (Kim & Park, 2019), as well as with the theory of automatization of lower-level processes. These models posit that during early writing acquisition, transcription skills demand substantial attentional and cognitive resources, limiting the availability of EFs to support OL. As transcription becomes automatized, cognitive resources are freed, allowing EFs to support higher-level processes.
Additionally, we note that prior studies examining EF effects on OL in kindergarten often assessed subcomponents (attention, working memory, inhibition, cognitive flexibility) separately. In contrast, our study treated EF as a unified composite measure, which may dilute the effect of specific subcomponents on OL. This methodological consideration helps explain why no direct EF effect on OL was observed in our sample.
MINOR ISSUES
- In pp.10 (lines 456-457) the sentence “Parents rated activity frequency on a scale from 1 (rarely/never) to 4 (almost daily)” is duplicated.
We have removed tis duplicated sentence.
- There is something missing/wrong in the following sentence “offers a novel perspective their synergistic enhancement of transcription skills”
Corrected, we have substituted this sentence for “offers a novel perspective on how they synergistically enhance transcription skills”
Final comment by authors:
We sincerely thank the reviewer for their thorough and constructive comments. Their insightful suggestions have significantly helped us improve the clarity, depth, and rigor of our manuscript. We greatly appreciate the time and effort invested in providing such detailed feedback, which has enhanced the overall quality of our work.
Reviewer 2 Report
Comments and Suggestions for Authors
See attached review

Author Response
I appreciate the opportunity to review the article titled ‘Developmental Trajectories of Transcription and Oral Language Skills in Kindergarten Students: The Influence of Executive Functions and Home Literacy Practices.’ This paper investigates the important components that inform children’s ability to write. I particularly appreciate the thoughtful review of the literature and the attention to details to help the field better understand the development of children’s writing skills.
Introduction
In general, the introduction is thorough and well written. I do think the introduction could be streamlined and in some places is repetitive.
- On lines 68 and 69 the sentences, “This recently developed model provides a new perspective on the main writing components. Transcription and oral language skills still play a crucial role in written composition,” confused me. I am unsure what model the authors are referring to and what the main components are.
We agree with this point; the initial sentence of this paragraph could be unclear. We have modified the initial sentence in order to clarify this part. See line 103.
- On lines 76 and 77, EF is defined a second time that is duplicative of the previous definition.
We have removed the duplicative information.
- On lines 207 and 208, it is not clear to me what the sentences, “Similarly, independent HLP, involving self-initiated writing activities, positively influence spelling Aram & Levin, 2001; Guo et al., 2021), handwriting (Guo et al., 2021; Puranik et al., 2018), and spontaneous writing (Puranik et al., 2018), indicate. Are the authors saying independent HLP is the same as dependent-formal HLP?
We would like to clarify that independent HLP are distinct from dependent-formal HLP. Independent HLP involve self-initiated writing activities carried out by the child without parental guidance, whereas dependent-formal HLP are structured literacy activities actively guided by parents. To make this distinction clearer, we have revised the text to explicitly indicate that independent HLP are autonomous and separate from dependent-formal practices. See line 245.
- Overall, the Streamline writing development at early grades: Theorical models section could be clearer and streamlined.
We have divided the introduction section by 2 parts in order to enhance comprehension: 1) Writing development at early grades; 2) Theorical models of writing. See line 56.
- On lines 233 and 234, the current literature review does not support the claim that HLPs influences child EF. Nor am I clear on how family SES is related to the association between HLPs and EF.
We agree that the original paragraph included information that may have extended beyond the direct aims of the study. Our intention was not to argue that HLP directly determine EF development, but rather to acknowledge, in line with prior work (e.g., Evans & Kim, 2013; Korucu et al., 2023), that children’s opportunities to engage in cognitively demanding literacy-related activities may contribute indirectly to EF growth. At the same time, we recognize that SES can influence the availability and quality of HLP, and thus potentially affect the final results. To avoid confusion, we have streamlined this section of the introduction. In the revised version, SES is now discussed primarily as a background factor that justifies its inclusion as a mediator in our analyses, while keeping the focus on HLP, EF, transcription, and oral language as the central constructs of this study. You can see this reduced section in line 252.
- On lines 238-240, It is not clear if the authors view EF as an external influence. I would argue EF is an internal process. Maybe the sentence is just not clear?
We agree that executive functions are internal cognitive processes rather than external influences. To clarify this distinction, we have revised the description of our study aims to explicitly differentiate internal processes (EF) from external factors (HLP) as influences on the development of transcription and oral language skills. See line 284.
- On line 251, the authors, again, do not explain how EF is “deeply shaped by socioeconomic context.”
Previous literature have demonstrated that lower-SES could influences EF development, however, we understand that all this literature could be more irrelevant with the objectives of our current study. Therefore, we have clarified in the manuscript that socioeconomic status was included as a control variable because research indicates that children from lower-SES backgrounds may have fewer opportunities for cognitively demanding interactions at home, which can affect the children’s development. See line 286.
- On line 258, I do not understand how children “develop executive functions through structured, language-rich exchanges (Evans & Kim, 2013).” This sentence appears to be a lynchpin to the authors’ argument for investigating these specific research questions. However, I do not think it is needed. The introduction does a very clear job of explaining how EF is a component to writing abilities. I do not think a link to from child language to EF is necessary. Similarly, I don’t think the link between SES and EF is clear or really needed. I think a simplified, straight-forward argument for why both HLPs and EF independently contribute to the development of writing would improve the introduction.
We agree that the sentence in question was not essential to the main argument of the study. Accordingly, we have removed the sentence linking child language and SES to the development of executive functions. The introduction now provides a more streamlined rationale, focusing directly on how both home literacy practices and executive functions contribute to early writing development.
Study Design, Analyses, and Results
Unfortunately, I think there is a fatal flaw in the design of this study as it is concurrently conceived. The fact that EF and HLPs were assessed at the end of study makes it difficult to understand how they would be related to the intercept (fall of the previous year) and growth in writing skills over the kindergarten school year. This is especially true given that the literature review discusses the bidirectional and reciprocal growth in these skills that we would anticipate
during a school year. I would recommend that a future draft of this paper consider concurrent associations at a single timepoint.
We fully acknowledge the limitations that EF and HLP were assessed only at the end of the school year. We have now explicitly mentioned this limitation in the Discussion section, emphasizing that the results should be interpreted with caution regarding causal inference (see line 797). While we agree that longitudinal measurement of EF and HLP would strengthen causal interpretations, the primary goal of the current study was to investigate the contribution of these constructs to the growth trajectories of transcription and oral language skills over the kindergarten year. The hierarchical linear modeling approach allows us to examine these associations in terms of intercepts and slopes, providing insight into potential influences even when predictors are measured at a single time point, which is common practice in similar early childhood research (e.g., Kim & Park, 2019; Berninger & Winn, 2006). Future studies could extend this design by assessing EF and HLP at multiple points to fully capture reciprocal developmental processes.
- I wonder if the authors considered the use of SEM given the multiple indicators of their constructs. If CFAs are used more information is needed about how the individual tasks were scored and then combined.
We appreciate the reviewer’s comment regarding the use of CFA for the EF and HLP constructs. We would like to clarify that both EF and HLP latent structures were indeed validated using CFA prior to calculating composite scores for subsequent HLM analyses. Specifically, for EF, the CFA confirmed good internal consistency and model fit (Ω = .723; χ² = 3.34, p > .05; CFI = 1.00; RMSEA = .00), and composite EF scores were computed as the sum of the observed measures. For HLP, a three-factor CFA (DF_HLP, DI_HLP, I_HLP) provided excellent model fit (Ω = .904; adjusted χ² = 28.70, p = .93; CFI = .96; RMSEA = .05), and scores for each HLP type were computed as the sum of their items. These composite scores were then used as predictors in all hierarchical linear models reported in the manuscript. We reported this information the “Executive function structure” and “Home literacy practices structure” sections (see line 552 and 561).
- I question the ability to estimate both person and school random effects given the small sample size. On average, how many children shared the same school? How many schools were in the sample?
The study included 7 schools with the following distribution of students: 29, 28, 7, 37, 13, 6, and 11 students per school. Although some schools have small samples, the inclusion of school as a random effect is justified by the hierarchical modeling procedure. Prior to fitting the final models, we tested the significance of random effects in Phase 1: Random Effects and Phase 2: Baseline Model.
For transcription skills (TR), the final random-effects structure included student-level intercept and slope, and school-level intercept only.
For oral language (OL), student-level intercept and slope were included, but school-level effects were not significant.
This approach follows standard recommendations for HLM, where the inclusion of random effects is based on empirical significance rather than solely on the number of units (Gelman & Hill, 2007; Snijders & Bosker, 2011). This information is added in lines 606 and 657.
- I am a bit confused with the focus on model fit, given that the RQs are focused on the contributions of individual predictors, not on testing or comparing different models.
Actually, we would like to clarify that the focus on model fit in our HLM analyses is not intended to compare theoretical models, but rather to identify the optimal random-effects structure prior to examining the influence of fixed predictors (HLP, EF, SES) on both intercepts and slopes. The modeling process was systematic and followed standard procedures for hierarchical linear modeling (this is an example for TR):
Phase 1 – Random Effects:
We first estimated the intraclass correlation (ICC) for student and school levels. Random effects were retained if ICC > 0.10, in line with the literature (Snijders & Bosker, 2011).
Phase 2 – Baseline Model (Random Effects Selection):
For Transcription (TR), the following random effects were compared:
- Student-level intercept only
- Student-level slope only
- Student-level intercept and slope
-> between these three models, the model c. was the best.
- School-level intercept only
- School-level slope only
- School-level intercept and slope
-> between these three models, the model d. was the best.
Subsequently, the two best-performing models (student-level and school-level) and a combined model of both, were compared to identify the final baseline random-effects model:
- Student-level intercept and slope
- School-level intercept only
- Combined model of c. and d.
-> between these three models, the model g. was the best, so this model will be the baseline model for the inclusion of fixed effects in phase 3.
Phase 3 – Fixed Effects Inclusion:
Once the baseline random-effects model was established, fixed effects (HLP and EF) were added. We systematically compared models including fixed effects on the intercept only versus intercept and slope to determine where each predictor exerted its influence.
In summary, model fit statistics were used to guide the selection of an empirically supported random-effects structure, ensuring that the final HLM provides reliable estimates of fixed effects without overfitting. This approach is standard in longitudinal hierarchical analyses (e.g., Singer & Willett, 2003) and is essential when evaluating contributions of individual predictors to growth trajectories. We understand that this process could be confuse, so we have clarified this step-by-step procedure in the Data Analysis section of the manuscript (see line 258).
Final comment by authors:
We sincerely thank the reviewer for the careful reading of our manuscript and for the thoughtful and constructive comments. We believe that the revisions and clarifications made in response to these suggestions have strengthened the manuscript, improving both its clarity and methodological rigor. We hope that these changes adequately address the reviewer’s concerns and meet the high standards of the journal.